# The pore structure of *Clostridium perfringens* epsilon toxin

Christos G. Savva[1], Alice R. Clark[2], Claire E. Naylor[3], Michel R. Popoff[4], David S. Moss[5], Ajit K. Basak[5], Richard W. Titball[6] & Monika Bokori-Brown [6]

Epsilon toxin (Etx), a potent pore forming toxin (PFT) produced by *Clostridium perfringens*, is responsible for the pathogenesis of enterotoxaemia of ruminants and has been suggested to play a role in multiple sclerosis in humans. Etx is a member of the aerolysin family of β-PFTs (aβ-PFTs). While the Etx soluble monomer structure was solved in 2004, Etx pore structure has remained elusive due to the difficulty of isolating the pore complex. Here we show the cryo-electron microscopy structure of Etx pore assembled on the membrane of susceptible cells. The pore structure explains important mutant phenotypes and suggests that the double β-barrel, a common feature of the aβ-PFTs, may be an important structural element in driving efficient pore formation. These insights provide the framework for the development of novel therapeutics to prevent human and animal infections, and are relevant for nano-biotechnology applications.

[1] Leicester Institute of Structural and Chemical Biology, Department of Molecular and Cell Biology, University of Leicester, Lancaster Road, Leicester LE1 7HB, UK. [2] Faculty of Science and Engineering, University of Wolverhampton, Wulfruna Street, Wolverhampton WV1 1LY, UK. [3] Molecular Dimensions, Willie Snaith Road, Newmarket CB8 7SQ, UK. [4] Bactéries Anaérobies et Toxines, Institut Pasteur, 25-28 Rue du Docteur Roux, 75724 Paris Cedex 15, France. [5] Department of Biological Sciences, Birkbeck College, Malet Street, London WC1E 7HX, UK. [6] College of Life and Environmental Sciences, University of Exeter, Stocker Road, Exeter EX4 4QD, UK. Correspondence and requests for materials should be addressed to M.B-B. (email: m.bokori-brown@exeter.ac.uk)

Epsilon toxin (Etx) is a pore-forming toxin (PFT) produced by *Clostridium perfringens* type B and D strains and plays an important role in the pathogenesis of enterotoxaemia, a severe neurological disease of domestic ruminants, particularly sheep. More recently, Etx has been implicated in the development of multiple sclerosis (MS) in humans[1–3].

Etx is typically secreted by the bacterium in the gut as a relatively inactive, water-soluble monomer, called protoxin (P-Etx), with an estimated molecular weight of 32.9 kDa[4]. The protoxin is converted to the active, mature toxin after carboxy- and amino-termini peptides are removed by proteolytic cleavage in the gut, either by digestive proteases of the host, such as trypsin, or protease produced by *C. perfringens*[5]. Toxin activation generates a more acidic protein (isoelectric point of 5.4 vs. 8.3) that is nearly 1000 times more toxic than its progenitor[6,7]. By a mechanism unknown, Etx crosses the gut wall, enters the blood stream and accumulates preferentially in the brain and kidneys[8,9].

Etx is a member of the aerolysin family of β-PFTs (aβ-PFTs). Members of the aβ-PFTs are found in all kingdoms of life and can be used for either attack or defence[10,11]. Other bacterial toxins from the aβ-PFTs that play a role in disease include aerolysin from the human pathogen *Aeromonas hydrophila*[12], enterotoxin (CPE) from *C. perfringens*[13], α-toxin from *Clostridium septicum*[14] and monalysin from *Pseudomonas entomophila*[15]. Other members of the aβ-PFTs serve in defence, such as lysenin produced by immune cells of the earthworm *Eisenia fetida*[16], assist in prey digestion, such as hydralysins from *Cnidaria*[17], or have cytocidal activity against human cancer cells, such as parasporin-2 from *Bacillus thuringiensis*[18].

As evidenced by their crystal structures, members of the aβ-PFTs share an overall similar domain arrangement[11]. They typically contain one or more receptor-binding domains (RBDs) and a pore-forming module (PFM), a well-conserved structural element of the aβ-PFTs[11,19]. The PFM, rich in serine and threonine residues, contains a flexible membrane insertion loop flanked by pre-insertion strands that refold into an amphipathic β-hairpin during pore formation to create the transmembrane β-barrel of the pore[20].

The crystal structure of the water-soluble, monomeric P-Etx was solved by Cole et al. in 2004[21] and revealed an elongated molecule composed mainly of β-sheets, with two strands traversing the entire molecule. The suggested RBD at the amino terminus contains a cluster of surface-exposed aromatic amino acids critical for cell binding[22,23], and the PFM at the carboxy terminus is suggested to play a role in oligomerization and pore formation[11,19]. The PFM also contains a C-terminal peptide (CTP), proteolytic removal of which is essential for toxin activation[6] as interaction of the CTP with the pre-insertion strands restricts their movement, thus preventing pore formation[20]. A glycan (β-octyl-glucoside (β-OG)) binding site has also recently been identified in the PFM, suggesting that Etx may have two distinct receptor-binding sites[22].

Etx is unique amongst aβ-PFTs as it shows high potency and high cell specificity. The 50% lethal dose ($LD_{50}$) in mice after intravenous administration of the toxin is reported to be 100 ng $kg^{-1}$ [24], which is equivalent to 7 μg per 70 kg human. This makes Etx the third most potent clostridial toxin after botulinum and tetanus neurotoxins[24]. For this reason, Etx is classified as a potential biological weapon in some countries and as a category B biological agent by the Centres for Disease Control and Prevention (CDC), the leading public health institute of the United States[25]. Among the many cell lines tested, only a few are susceptible to Etx. Most in vitro studies on Etx, including this study, have been performed using the Madin–Darby Canine Kidney (MDCK) cell line as this cell line is the most sensitive to the toxin[26–29]. The cell specificity of Etx suggests that the toxin binds to a specific receptor in the membrane of target cells and this is likely to explain why the toxin is over 1000 times more potent compared to other aβ-PFTs.

To date, two putative Etx receptors have been identified: the O-linked glycoprotein hepatitis A virus cellular receptor 1 (HAVCR1)[23] and the tetraspan membrane proteolipid myelin and lymphocyte protein (MAL)[2]. The molecular basis for the interaction of these putative receptors with Etx remains unknown, although it is thought that the binding site for Etx on the glycoprotein HAVCR1 includes one of the sugar components, while in MAL the second extracellular loop is critical for binding and cytotoxicity of Etx. Etx can also interact with artificial lipid bilayers and form functional pores without the help of receptors, albeit with less efficiency[30–32].

Both inactive P-Etx and activated toxin can bind to cells, but only activated toxin with CTP removed can form heptameric pre-pores on the membrane surface in the next step of pore formation[33]. Finally, the heptamer, with a molecular mass of approximately 155–174 kDa[2,28,29,34], inserts into the membrane, which is triggered by significant structural rearrangements within the PFM of each protomer in the complex. Pore formation disrupts the membrane and is central to the damage caused to intoxicated cells, and consequently to the symptoms of disease, due to the leakage of water and small ions through the pores that may lead to cell death.

Whilst atomic resolution structures of the water-soluble, monomeric forms have been determined for many members of the aβ-PFTs over the past few decades, their corresponding pore structures have only recently been revealed. The lysenin pore was the first high-resolution structure of the aβ-PFTs[20,35]. A 3.9 Å structure of the aerolysin pre-pore and a 7.5 Å structure of aerolysin pore have also been determined[36].

The cryo-electron microscopy (cryo-EM) structure of Etx pore determined here provides further insights into the molecular details of structural rearrangements during pore formation of the aβ-PFTs. These results will have important implications for the development of novel therapeutics aimed at the functional disruption of pore formation to prevent human and animal diseases, and are relevant for nano-biotechnology applications.

## Results

**Isolation of the Etx pore**. The formation of biologically active oligomers has been a limiting factor for obtaining structural information of the pore form of Etx. Unlike aerolysin[37,38], Etx does not spontaneously oligomerize into sodium dodecyl sulfate (SDS)-resistant species when activated in aqueous buffer, and negatively stained EM samples do not reveal regular particles other than elongated monomers and small clumps (Supplementary Fig. 1a). Furthermore, efficient oligomerization cannot be promoted by the addition of liposomes of various lipid compositions or detergent micelles. The uncertainty in receptor identity combined with the high cell specificity of Etx prompted us to investigate if the pores could be assembled on susceptible cells and then purified for structural studies. We assembled Etx oligomers that were biologically active towards Super Dome cells, a clone of MDCK cells that forms domes that are approximately five times the area of MDCK cells, thus exhibiting exceptional sensitivity to Etx[39]. The cytotoxic activity of trypsin-activated recombinant wild-type Etx towards Super Dome cells was measured using a lactate dehydrogenase (LDH) assay. The average dose of recombinant Etx required to kill 50% of Super Dome cells ($CT_{50}$) was $2.47 \pm 0.47$ nM (SEM) or $78.27 \pm 14.89$ ng ml$^{-1}$ (SEM) (Fig. 1a). Etx monomers oligomerized and formed SDS-resistant pores of an apparent molecular weight of ~155 kDa on the membrane of Super Dome cells (Fig. 1b), consistent

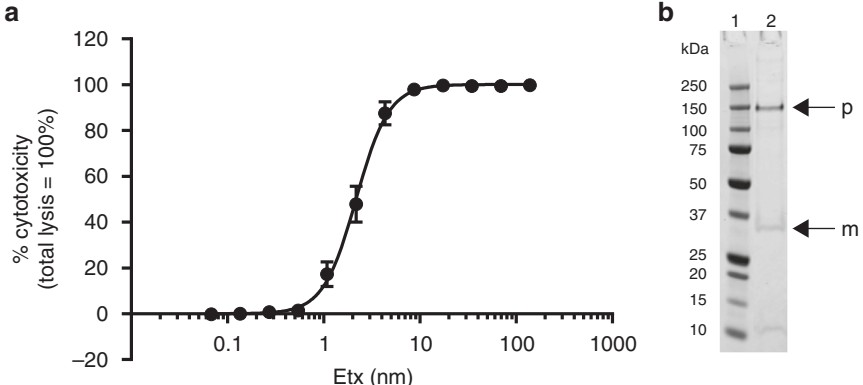

**Fig. 1** Cytotoxic activity of wild-type epsilon toxin (Etx) towards Super Dome cells. **a** The cytotoxic activity of trypsin-activated recombinant wild-type Etx towards Super Dome cells was determined by measuring the release of lactate dehydrogenase (LDH) from lysed cells. Results were normalized to the signal from cells treated with Dulbecco's phosphate-buffered saline (DPBS) only (0% lysis) and cells treated with 0.9% (v/v) Triton X-100 (100% lysis). Results are presented as the mean of triplicate assays, each performed in triplicate (±SEM). **b** Purified Etx pore was separated by sodium dodecyl sulfate-polyacrylamide gel electrophoresis (SDS-PAGE) and visualized by Coomassie staining. Arrows indicate the positions of Etx pore (p) and Etx monomer (m). Source data are provided as a Source Data file

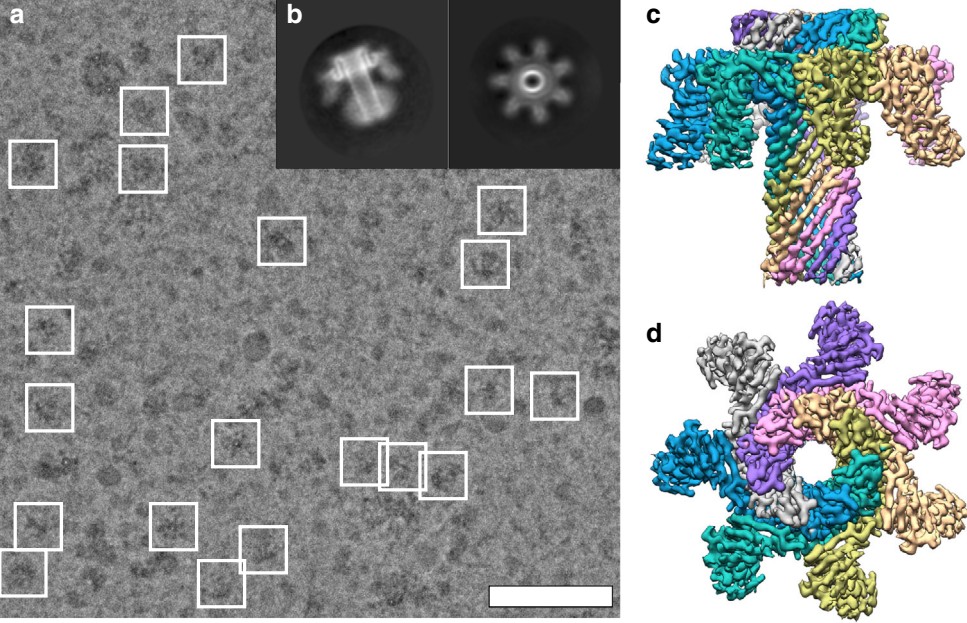

**Fig. 2** Cryo-electron microscopy (EM) of epsilon toxin (Etx) pores using the Volta phase plate. **a** Electron micrograph of a typical field of view of isolated Etx pores. Particles that were included in the final 3D map are highlighted. **b** Characteristic 2D class averages. Sharpened cryo-EM map of the pore coloured by monomer as visualized from the side (**c**) and extracellular (**d**) views. Scale bar corresponds to 50 nm

with previous studies[2,28,29,34]. Subsequently, the complexes were solubilized in detergent and treated as single particles for cryo-EM analysis. Solubility screening using ultracentrifugation identified $n$-dodecyl-β-D-maltoside (DDM) as a suitable detergent, which has previously been successfully used to solubilize lysenin and Etx pores[20,34,35] (Supplementary Fig. 2).

**Structural determination of the Etx pore**. When observed using transmission EM, the isolated Etx material appeared heterogeneous. The extremely low yields (~10 μg from $6 \times 10^8$ cells) meant that we could not pursue additional clean-up steps, such as size-exclusion chromatography. This heterogeneity hindered particle identification and selection using manual or automated picking procedures (Supplementary Figs. 3, 4). Despite this, an initial dataset of 613 micrographs resulted in ~14,000 particles after 2D and 3D classification clean-up steps and produced a map that reached 4.6 Å, at which resolution the β-strands just begin to separate (Supplementary Fig. 3, inset). We reasoned that the use of a Volta phase plate (VPP) would be advantageous for two reasons. First, the contrast of the protein component would increase, aiding particle picking and alignment. Second, the number of particles required to achieve a high-resolution structure could decrease as compared to conventional defocused data[40]. Therefore, we proceeded to image the exact same grid used before with the VPP. Although the samples still appeared heterogeneous (Fig. 2a), approximately two times as many particles remained after clean-up steps as compared to the non-VPP dataset from a dataset that was only 25% larger. This resulted in a final map at an overall resolution of 3.2 Å (Fig. 2c, d and

Supplementary Fig. 5). Estimation of the *B*-factors for both datasets indicated that the phase plate data was indeed of higher quality when comparing the same number of particles used in a reconstruction (Supplementary Fig. 6).

**Architecture of the Etx pore**. The atomic model of the Etx pore is shown in Fig. 3a, b. The overall architecture of the Etx pore resembles that of its structural homologues lysenin[20,35] and aerolysin[36], with a β-barrel that spans the height of the molecule, a motif that so far seems specific for the αβ-PFTs. The Etx pore also lacks a vestibule that is found in the α-hemolysin family of β-PFTs where the β-barrel begins close to the membrane-spanning region (Supplementary Fig. 7a, d, respectively). The Etx assembly measures ~120 Å across and ~98 Å in height. The lumen of the β-barrel measures ~24 Å (Cα–Cα) at its widest, although local restrictions are present (see below).

Each protomer in the pore structure can be divided into three domains (Fig. 3a, b): the β-hairpin domain (blue), the cap domain (yellow) and the RBD (green). The β-hairpin domain (residues 95–173) creates the inner β-barrel, which begins at the top of the pore and runs through the entire length of the assembly, including the membrane-spanning region. The cap domain (residues 63–94, 174–198 and 235–260) at the extracellular side, also referred to as the collar[35], includes an outer β-barrel that surrounds the membrane-distal region of the inner β-barrel, and thus contributes to the highly conserved, concentric double β-barrel (DBB) fold, a common feature of the αβ-PFTs[41]. The RBD (residues 1–62 and 199–234) is unique to each αβ-PFT and determines the binding specificity of the toxin.

**The β-hairpin domain**. As has been shown previously[20], members of the αβ-PFTs form β-barrels that span the height of their pores. This domain includes the insertion loops that have been hypothesized to form the membrane-spanning regions (Etx residues 123–148)[21,42,43]. As in lysenin, in fact the β-barrel recruits polypeptide from the flanking β-strands on either side of

the insertion loop so that the β-hairpin domain actually comprises residues 95–173. The lumen of the pore is mostly lined with serine and threonine residues, yet another characteristic of the aerolysin family, but asparagine and glutamine residues are also present. Additionally, a number of salt bridges are present (Supplementary Fig. 8c–e). Forming intra-chain bridges are Lys99-Asp168 and Lys108-Glu159. Lys130 with Glu137 form an inter-chain salt bridge, which stabilizes the β-barrel near the intracellular side of the pore in addition to the hydrogen bonding network. The overall charge distribution in the lumen is therefore neutral (Supplementary Fig. 9b). At its narrowest point the diameter of the lumen is constricted to ~12 Å by the seven Lys108 residues, which protrude into the lumen (Supplementary Figs. 8a, 9b, rectangle). The structure corroborates studies that show Etx has only a slight preference for anions[31,32]. The β-turn at the intracellular side is formed by Thr132, Val133, Pro134 and Phe135. Aerolysin and Etx are conserved in this region. Pro134 is equivalent to Pro248 in aerolysin and, as in aerolysin, the β-turn main chain protrudes sideways, forming a rivet to anchor the barrel at the intracellular side (Fig. 3c)[44].

The external surface of the Etx inner β-barrel reveals a number of interesting features (Fig. 3c). In the intracellular side of the membrane two aromatic residues, Phe131 and Phe135, form the usual aromatic belt found in β-barrel transmembrane regions. However, in the extracellular side of the membrane there are two aromatic belts. One is formed by Tyr146 ~30 Å from the tip, and the other formed by His119 (H106 in Etx without propeptide[45]) and Phe148 ~35 Å from the tip.

Examination of the unfiltered maps indicates that the detergent micelle density extends above His119 and Phe148, suggesting that these residues will be interacting with the membrane (Supplementary Fig. 10). Therefore, we propose that the membrane-inserted region in the Etx pore comprises residues 119–148, close to the original prediction[21]. Interestingly, histidine residues were also found in the detergent micelle region of lysenin above the standard aromatic belt[20]. Towards the membrane-distal part of the inner β-barrel, His162

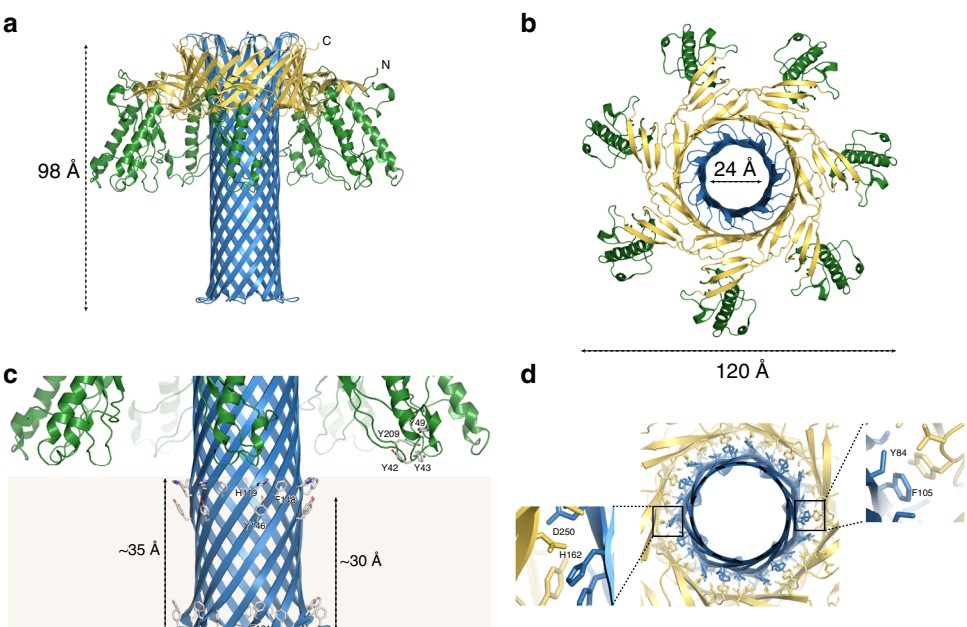

**Fig. 3** Atomic model of the epsilon toxin (Etx) pore. Cartoon representation of the heptameric Etx model with height, width and lumen dimensions indicated (**a**, **b**). The monomers are coloured by domain: green—receptor-binding domain (RBD); blue—β-hairpin domain; yellow—cap domain. **c** Close-up view of the inner β-barrel and membrane-spanning region. Residues identified in previous studies as important to receptor binding in the RBD, as well as residues forming the aromatic belts, are highlighted in stick representation. The approximate boundaries of the lipid bilayer are coloured light brown. **d** Close-up view of the double β-barrel (DBB) fold viewed from the membrane side with interacting residues shown in stick representation

(H149 in Etx without propeptide) is another important residue for Etx activity[22,45]. H162 (Nδ1) is within hydrogen bonding distance of D250 (Oδ2), which lies in the cap domain of the same chain (Fig. 3d and Supplementary Fig. 8a).

**The RBD.** This region undergoes minimal conformational changes in the transition from the water-soluble form, as in lysenin[20]. Unique to Etx, and unlike lysenin and aerolysin, the individual RBDs are well separated and do not interact with their neighbours (Fig. 3b and Supplementary Fig. 7a–c), and thus do not contribute to any oligomerization surfaces. The lack of physical constraints in the RBD could infer higher flexibility in receptor selection and binding, and in part explains why the resolution of this domain is lower than that of the other domains (Supplementary Fig. 5). Overall, the RBD in the oligomer overlaps well with the corresponding region of the crystal structure of Etx in its monomeric form (PDB ID: 1UYJ) (Supplementary Fig. 11), with a root-mean-square deviation of 1.7 Å (Cα–Cα). A number of tyrosine residues (Tyr 42, 43, 49 and 209; corresponding to Tyr 29, 30, 36 and 196 in refs. [22,46]), which were identified in earlier studies as important in receptor binding[22,46], are located directly above the membrane in a position that would facilitate interaction with a receptor (Fig. 3c). As shown in Supplementary Fig. 9a, d, the tips of the RBD are predominantly hydrophobic.

**The cap domain.** The cap domain links the RBD to the β-hairpin domain and undergoes significant rearrangement in the pore form, as discussed below. In the pore conformation it contributes the outer β-barrel to the DBB fold (Fig. 3d). In aerolysin, the DBB fold has been shown to be responsible for the SDS-resistant character of aerolysin[36]. In Etx the DBB fold is stabilized by hydrophobic interactions between the inner and outer β-barrel that includes valines, leucines, isoleucines and offset π–π stacking between Tyr84 and Phe105 residues, as well as some hydrogen bonds, including H162 and D250 mentioned above (Fig. 3d and Supplementary Fig. 8a, b).

To investigate the importance of the H162–D250 interaction, we constructed D250A and H162A mutants. Etx-H162A (corresponding to H149A in ref. [45]) has previously been reported to have reduced, but not abolished, toxicity[45], and the crystal structure of P-Etx-H162A closely resembles that of wild-type P-Etx[22], while purified recombinant Etx-D250A is completely inactive towards Super Dome cells, as shown in Supplementary Fig. 12a, and thermal stability assays indicate that P-Etx-D250A

behaves similarly to wild-type P-Etx (Supplementary Fig. 12b), suggesting that P-Etx-D250A is correctly folded. Examination of D250 and H162 in the monomeric Etx crystal structure indicates that D250 and H162 are not involved in any interactions, suggesting that the importance of these residues lies in the H162–D250 hydrogen bond formed in the oligomer. SDS-PAGE analysis of oligomer formation revealed that both Etx-D250A and Etx-H162A produce SDS-resistant oligomers of the same apparent molecular weight as wild-type Etx (~155 kDa, consistent with previous reports[2,28,29,34]) (Supplementary Fig. 13, lanes 11–13). When Super Dome cells were incubated with activated wild-type Etx, Etx-D250A or Etx-H162A, oligomers were formed with a mean oligomer yield of 5.09 ± 2.62% (SEM) for Etx-D250A (Supplementary Fig. 13b), albeit in a concentration-dependent manner (Supplementary Fig. 14), and with a mean oligomer yield of 44.54 ± 10.25% (SEM) for Etx-H162A relative to wild-type Etx (Supplementary Fig. 13b). These results suggest that while D250A oligomers resembling the wild-type ones, either in a pre-pore or post-pre-pore state, can assemble, they cannot produce pores that have a cytotoxic effect on cells (Supplementary Fig. 12a), even at the highest concentration of 6 μM tested (2426 times the $CT_{50}$ dose of wild-type Etx). Further analysis of oligomer formation by native-PAGE revealed a much larger complex of ~624 kDa relative to the SDS-resistant oligomers of ~155 kDa when Super Dome cells were incubated with activated wild-type Etx (Supplementary Fig. 15, lane 10). The same complex, albeit at lower intensity, was also observed when cells were incubated with activated Etx-H162A (Supplementary Fig. 15, lane 12), but was not detectable when cells were incubated with activated Etx-D250A (Supplementary Fig. 15, lane 11), providing further evidence that D250A oligomers detected by SDS-PAGE may not correspond to pores. Both D250A and H162A mutants retain their ability to bind to Super Dome cells (Supplementary Fig. 13a, lanes 8–13, and Supplementary Fig. 15a, lanes 7–12), suggesting that residues D250 and H162 are important for events downstream of receptor binding. Similar to activated wild-type Etx, negative stain EM did not reveal regular particles other than elongated monomers and small clumps for activated Etx-D250A and Etx-H162A (Supplementary Fig. 1).

**Structural rearrangements during pore formation.** Comparison of the crystal structure of Etx monomer to its membrane-inserted form illustrates the large conformational changes that occur upon pore formation (Fig. 4 and Supplementary Movie 1). The cap

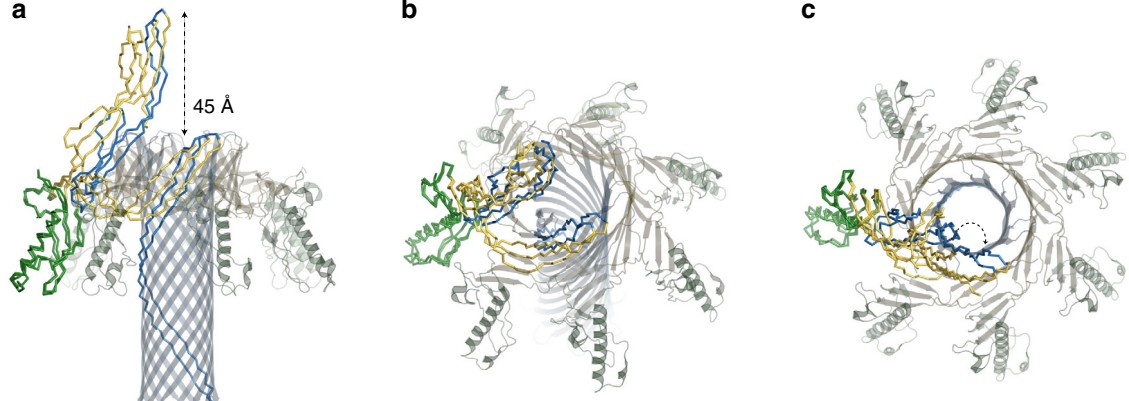

**Fig. 4** Conformational changes accompanying pore formation. Wire representation of epsilon toxin (Etx) monomers in the water-soluble and membrane-inserted state coloured by domain as in Fig. 3 and viewed from different angles (**a–c**). The two models were superimposed by their receptor-binding domain (RBD), which remains mostly unchanged during the transition. The cap domain collapses by 45 Å allowing the pre-insertion strands to get closer to the membrane and form the inner β-barrel. The heptameric pore is also shown for context. Curved arrow in **c** illustrates outward movement of the cap domain in the membrane-inserted form

domain moves downward towards the membrane by 45 Å creating a kink in the straight cap domain. In turn, the β-hairpin domain, consisting of the hypothetical insertion loop and the pre-insertion strands, as for lysenin[20], unfolds to create the inner β-barrel that spans the entire length of the Etx pore. In contrast to lysenin, the cap domain in Etx moves slightly away from the inner β-barrel rather than towards it.

To further understand how Etx assembles into an oligomer and the rearrangements required for transition to the pore state, we constructed a hypothetical pre-pore by superimposing the crystal structure monomer to the RBD of the pore structure and imposing 7-fold symmetry, similar to hypothetical pre-pores constructed previously for lysenin[20,35]. This allowed us to observe some of the displacements that have to occur prior to pre-pore formation (Supplementary Fig. 16). Calculations of molecular clashes (inter-atom overlap >0.6 Å) highlight the potential regions that would have to rearrange upon oligomerization. It is clear from these clashes that the CTP, as expected, severely obstructs oligomerization, and the pre-pore model illustrates why this region must be removed prior to monomer association. In addition to the CTP, other significant clashes occur near the oligomerization surfaces on either side of a monomer, indicating again that some rearrangement will have to happen. Finally, the hypothetical insertion loop or tongue also creates clashes upon monomer association and, as described previously for lysenin[35], the displacement of the tongue region may be the driving force for the pre-insertion strands to unfold and the formation of the inner β-barrel.

## Discussion

The pore form of Etx has long eluded structural determination, mostly owing to the difficulty of forming biologically active Etx oligomers in liposomes of various lipid compositions or detergent micelles without a receptor, which is unknown. Thus, the only viable way to study the pore form of Etx was to isolate the complexes directly from cells known to be susceptible to Etx. In this study, the use of Super Dome cells, reported to have 3- to 78-fold increased sensitivity to Etx relative to MDCK cells[39], combined with the use of increased concentration of DDM (2% (w/v)) relative to 0.05% (w/v) used in ref. [34] maximized pore yield for subsequent purification, and thus were critical for structural determination of the Etx pore complex. Furthermore, due to the very small quantities of material that can be isolated from the membrane of cultured cells and the heterogeneity of the purified samples, cryo-EM technique was the only structural method that could be employed. Use of the VPP proved essential in our studies to achieve high resolution, owing to the challenging sample rather than its molecular mass (~225 kDa). First, particles could be identified by the automatic and manual picking procedures more efficiently, resulting in more particles (Supplementary Fig. 4). Second, the overall quality of the data, as judged by B-factor estimation from both datasets, was higher for the VPP data, meaning we could achieve higher resolution than with the non-VPP data, given the number of particles (Supplementary Fig. 6). While this manuscript was under review, another cryo-EM study of a relatively large complex (~500 kDa) also benefited by the use of the VPP[47], suggesting that VPP use could be advantageous for certain specimens that are not size limited. The resulting map at 3.2 Å resolution allowed an unambiguous model to be built for Etx pore.

A number of interesting mutants of Etx have been studied in the past whilst trying to understand how Etx transitions from the water-soluble to the pore form. Mutating H149 (H162 in our model) to alanine resulted in a significant decrease in activity of the recombinant toxin both in vitro and in vivo[45]. However,

substitution with serine at position 162 does not abolish lethal activity[45], suggesting that H162A is less active not because of removal of the bulky imidazole side chain, but possibly due to the loss of a hydrogen bond donor at this position. H162, located in the membrane-distal part of the inner β-barrel, has previously been equated to aerolysin Y221, and recent studies proposed that, similar to the aerolysin Y221G mutant, the Etx-H162A mutant may prevent the β-hairpin domain involved in pore formation from unfolding[19,41]. However, unlike the aerolysin Y221G mutant, the Etx-H162A mutant did not reveal water-soluble, heptameric pre-pore particles by negative stain EM (Supplementary Fig. 1b).

Examination of the Etx pore structure reveals that H162 may also play a role in stabilizing the DBB fold by forming a hydrogen bond with D250 in the cap domain. This interaction will stabilize the DBB fold further in addition to the hydrophobic interactions and make it more rigid. In agreement with this hypothesis, substitution of aspartate with alanine at position 250 results in a completely inactive Etx that displays ~95% reduction in SDS-resistant oligomers relative to wild-type Etx. The reduced oligomerization of Etx-D250A and its complete inactivity towards Super Dome cells could be a result of either oligomer instability or a reduction in oligomer formation. Since D250 does not participate in any inter-chain interactions in the oligomer, we cannot explain the latter. The abolished cytotoxicity of Etx-D250A suggests that the SDS-resistant oligomers resembling the wild-type ones may correspond to pre-pores or post pre-pores. This may explain why Etx-D250A oligomers resembling the wild-type ones were not observed by native-PAGE immunoblot. Pre-pores or post-pre-pores are expected to migrate differently to pores on native-PAGE as they will lack any associated detergents or lipids.

Further evidence for the importance of the DBB stability in pore formation also comes from previous mutagenesis studies. Tyrosine 84 in the cap domain and F105 in the inner β-barrel have previously been identified as important to Etx function[48]. Y84A completely abolishes pore formation and is only recoverable by substitution to another aromatic residue, whilst F105A has reduced cytotoxic activity towards MDCK cells at low toxin concentrations[48]. In the pore structure, these two residues form a π-π stacking interaction (Fig. 3d and Supplementary Fig. 8b). Similar to D250A and H162A mutants, Y84A and F105A mutants also retain their ability to bind to MDCK cells.

Based on previous studies and the pore structure presented here, we propose that the stability of the DBB fold may be important in coupling the conformational changes that occur in the cap domain (the vertical collapse by 45 Å) to the insertion of the inner β-barrel into the membrane. By stabilizing the top end of the inner β-barrel as it is being formed by the unfolding of the β-hairpin domain, the insertion into the membrane will be analogous to a drill, which is forced into the ground by applying force at the top. Similarly, the RBDs with their hydrophobic tips may keep the complex anchored to the membrane during membrane insertion, analogous to the stabilizing feet of the drill, by either an interaction with the receptor or with the lipid bilayer itself. The pre-pore structure of aerolysin by cryo-EM[36] showed that the DBB is already formed at this stage and is a pre-requisite to insertion of the inner β-barrel into the membrane.

Monomeric P-Etx has previously been crystallized bound to β-OG in the PFM of the carboxy terminus[22]. In the pore form of Etx, the β-OG binding site is disrupted and the interacting residues (V85, F105 and T106, corresponding to V72, F92 and T93 in[22]) are scattered amongst the inner β-barrel and the cap domain. Similar to aerolysin[49], in order for Etx to cross the thick layer of glycocalyx covering the apical side of the plasma membrane of gut epithelial cells, Etx may initially bind to surface oligosaccharides with low affinity, possibly through the β-OG

binding site at the carboxy terminus, before it binds with high affinity to its receptor on the membrane surface through the RBD at the amino terminus. This hypothesis is supported by the finding that polarized epithelial cells are more sensitive to Etx from their apical side, which is more abundant in oligosaccharides, than from their basolateral side[29,50]. Binding of Etx to a double receptor in the nervous system has also been suggested previously[9].

Insights into the molecular details of structural rearrangements and monomer–monomer interactions established upon pore formation can be applied to structure-based drug design where high-affinity compounds are identified to block pore formation, thus fighting bacterial infections. Further studies on Etx will have important implications for developing novel therapeutics aimed to disrupt pore formation, thus preventing serious human and animal infections, and are relevant for nano-biotechnology applications.

## Methods

**Materials.** Chemicals were purchased from Sigma, UK, unless otherwise stated. The polyclonal antibody against Etx mutant Y30A-Y196A was raised in rabbits as described in ref. [51]. After electrophoretic separation purified Etx pore was detected by SimplyBlue staining (Invitrogen Ltd, Paisley, UK). Protein concentrations were determined using the Pierce™ BCA Protein Assay Kit (Fisher Scientific Ltd, Loughborough, UK). Synthetic oligonucleotides were sourced from Eurofins Genomics, Germany.

**Recombinant protein production and purification.** The gene, *etxB*, encoding epsilon protoxin (P-Etx) was PCR amplified using synthetic oligonucleotides Etx-*Nco*I-F and Etx-*Xho*I-R (Supplementary Table 1) designed to bear restriction enzyme sites *Nco*I and *Xho*I, respectively, to facilitate directional cloning into the expression vector pHis-Parallel1 (GenBank: AF097413.1). This fused the amino-terminal end of P-Etx without the 14 amino-terminal residues (KEISNTVS-NEMSKK) to the 26 amino acid residue-long peptide (MSYYHHHHHHDY-DIPTTENLYFQGAM), which contains an amino-terminal polyhistidine (6× His) affinity tag upstream of a spacer region (DYDIPTT) and the rTEV protease cleavage site (ENLYFQG). The fidelity of PCR amplification was verified by automated DNA sequencing (Eurofins Genomics, Germany). The recombinant plasmid expressing P-Etx is termed pHis-Parallel1-P-Etx. Amino acid numbering corresponds to P-Etx with the 13 amino acids amino-terminal peptide sequence, unless otherwise stated.

For expression of P-Etx, recombinant plasmid pHis-Parallel1-P-Etx was transformed into *E. coli* Rosetta 2 (DE3) cells (Merck, Darmstadt, Germany) and expression of P-Etx was induced using the autoinduction system as described before[22] but with modifications. In brief, cells (100 ml) were grown at ACDP/ACGM (Advisory Committee on Dangerous Pathogens/Advisory Committee on Genetic Modification) containment level 3 in ZYM-5052-autoinducing medium supplemented with 50 µg ml⁻¹ carbenicillin and 34 µg ml⁻¹ chloramphenicol and cultured at 37 °C for 3 h at 300 rpm, then for a further 24 h at 20 °C at 300 rpm. Cells were harvested by centrifugation at 10,000× *g* for 10 min at 4 °C and the cell pellet was lysed by 10 ml BugBuster™ Protein Extraction Reagent (Merck, Darmstadt, Germany) containing rlysozyme™ (0.5 µl at 30 KU µl⁻¹) (Merck, Darmstadt, Germany) and Benzonase® Nuclease (10 µl at 25 U µl⁻¹) (Merck, Darmstadt, Germany). The cell suspension was incubated on a rotating mixer for 25 min at room temperature and centrifuged at 16,000× *g* for 20 min at 4 °C to separate soluble and insoluble fractions. The supernatant was loaded onto His GraviTrap columns (GE Healthcare Life Sciences, Little Chalfont, UK) following the manufacturer's guidelines. In brief, His-tagged P-Etx was bound to the affinity column using a buffer composed of 20 mM sodium phosphate, 500 mM NaCl and 20 mM imidazole, pH 7.4. The column was washed with a buffer composed of 20 mM sodium phosphate, 500 mM NaCl and 60 mM imidazole, pH 7.4. Recombinant P-Etx was eluted in a buffer composed of 20 mM sodium phosphate, 500 mM NaCl and 500 mM imidazole, pH 7.4. All purification steps were carried out at 4 °C. For buffer exchange and sample clean-up, P-Etx-containing eluate was applied to a PD-10 Desalting Column (GE Healthcare Life Sciences, Little Chalfont, UK) and eluted in Dulbecco's phosphate-buffered saline (DPBS), pH 7.0–7.2 (Invitrogen). Protein concentrations were determined by BCA assay.

**Activation of P-Etx by trypsin.** Purified recombinant P-Etx was activated with trypsin, TPCK (L-(tosylamido-2-phenyl) ethyl chloromethyl ketone) treated from bovine pancreas, which removes the CTP sequence. Trypsin was prepared in DPBS and added to recombinant P-Etx at 1:100 (w/w) ratio and incubated at room temperature for 1 h. The reaction was stopped by adding trypsin inhibitor (0.66 mg per 1 mg trypsin) to the digest. Removal of the CTP sequence was assessed by SDS-PAGE.

**Site-directed mutagenesis.** Mutation D250A or H162A was introduced into wild-type recombinant Etx using primer pairs D250A-F and D250A-R or H162A-F and H162A-R, respectively (Supplementary Table 1) and the QuickChange Lightning Site-Directed Mutagenesis Kit (Agilent Technologies Inc, Santa Clara, CA, USA) according to the manufacturer's instructions. The presence of the intended mutation was verified by automated DNA sequencing (Eurofins Genomics, Germany).

**Thermostability assay.** Thermostability of purified recombinant P-Etx-D250A was assessed by mixing purified protein (0.25 mg ml⁻¹) with 240× SYPRO Orange protein gel stain. Fluorescence was monitored using a QuantStudio™ 6 Real-Time PCR System (Applied Biosystems, USA) with a 1% thermal gradient from 25 °C to 99 °C. The fluorescence data obtained was analysed using the Protein Thermal Shift Software (Applied Biosystems, USA) to calculate the melting temperature ($T_m$) using the Boltzmann method.

**Cell culture.** Super Dome (ATCC® CRL-2286™; American Type Culture Collection (ATCC), USA) cells, a canine epithelial-like cell line established by cloning from the MDCK (ATCC® CCL-34™) cell line, were routinely cultured in Dulbecco's modified Eagle's medium/Ham's F12 medium (DMEM/F12, HEPES; Life Technologies) with 2.5 mM L-glutamine, 15 mM HEPES, 0.5 mM sodium pyruvate and 1.2 g l⁻¹ sodium bicarbonate supplemented with 0.05 mM nonessential amino acids and 10% foetal bovine serum Gold (PAA, Pasching, Austria) at 37 °C in a humidified atmosphere of 95% air/5% CO₂. The culture medium was replaced every 2 to 3 days. Cells were routinely detached by incubation in trypsin/EDTA and split as appropriate (typically 1:5 dilutions). For dome formation, Super Dome cells were fed twice a day by a complete medium change.

**Cytotoxicity assay.** The cytotoxic activity of trypsin-activated Etx toward Super Dome cells was determined by measuring the amount of LDH released from the cytosol of lysed cells into the cell culture medium using the CytoTox 96 Non-radioactive Cytotoxicity Assay Kit (Promega, Southampton, UK) according to the manufacturer's protocol. In brief, a 2-fold dilution series of each activated toxin (ranging from 139 to 0.067 nM) was prepared in DPBS and added to cells seeded into 96-well plates (3 × 10⁴ cells per well). Following incubation at 37 °C for 3 h, the cell culture medium (50 µl) was harvested from cell monolayers, transferred to a fresh 96-well enzymatic assay plate and 50 µl of reconstituted substrate mix was added to each well. The plate was incubated for 30 min at room temperature, protected from light. Absorbance was measured at 490 nm using a Tecan Infinite 200 PRO Microplate Reader (Tecan Group Ltd, Switzerland). The absorbance values for each sample were normalized by subtracting the absorbance value obtained for the culture medium from untreated cells. The toxin dose required to kill 50% of the cell monolayer ($CT_{50}$) was determined by nonlinear regression analysis, fitting a variable slope log(dose) vs. response curve, constraining $F$ to a value of 50 ($\log CT_{50} = \log CTF - (1/\text{HillSlope}) \times \log(F/(100 - F))$).

**Etx pore assembly.** Super Dome cells were grown to confluency in 30× T175 flasks. Each flask of cells was incubated with 5 ml cell culture medium containing trypsin-activated Etx (25 µg ml⁻¹; 50× the average minimum dose required to lyse 50% cells) at 37 °C for 1 h to allow binding and pore formation. Lysed cells were centrifuged at 1500 × *g* for 15 min at 4 °C to pellet cellular debris and unbound Etx was removed by ultracentrifugation at 100,000 × *g* for 1 h at 4 °C using a Beckman Type 70Ti rotor. Solubilization of toxin complex from Super Dome cells was carried out by resuspending the pellet in DPBS (pH 7.0–7.2) containing 137 mM NaCl and 2% (w/v) *n*-dodecyl-β-D-maltoside (DDM) and incubation at 4 °C overnight, a modification of the method described by Shimada et al.[34] where the toxin complex from MDCK cells was solubilized by resuspending the pellet in 20 mM sodium phosphate buffer (pH 7.4) containing 0.05% (w/v) DDM and incubation at 25 °C for 1 h. After incubation, insoluble material was removed by ultracentrifugation at 75,000 × *g* for 45 min at 10 °C using a Beckman Type 70Ti rotor. Solubilized Etx oligomers were bound to Ni-NTA Agarose resin (Qiagen), washed with 2× 20 volumes of DPBS, 0.02% (w/v) DDM and eluted in three volumes of DPBS, 200 mM imidazole and 0.02% (w/v) DDM.

**Detergent screening.** To screen for a suitable detergent that could efficiently extract and maintain Etx pores soluble, Super Dome cell membrane was incubated with trypsin-activated Etx for 1 h at 37 °C and membrane incubations were distributed between nine tubes. After ultracentrifugation at 100,000 × *g* for 1 h at 4 °C using a TLA 120.1 rotor, each pellet was resuspended in DPBS containing 137 mM NaCl and detergent (1% (w/v) DDM, 1% (w/v) DM (*n*-decyl-β-D-maltopyranoside), 0.5% (w/v) C12E8, 1% (w/v) FC-12, 2% (w/v) β-OG, 2% (w/v) C8E4, 1% (w/v) CYMAL-5, 1% (w/v) LDAO (lauryldimethylamine-N-oxide) or 1% (w/v) LMNG (lauryl maltose neopentyl glycol)) and incubated overnight at 4 °C with occasional shaking. After incubation, 10 µl aliquots were removed from each tube for Western blot analysis (total protein in detergent) and insoluble material was removed by ultracentrifugation, as before. The supernatants, which contained solubilized Etx pore, were collected and 10 µl aliquots were removed from each tube to assess Etx pore solubilization by Western blot analysis using 6×-His Tag

Monoclonal Antibody (4A12E4) (Invitrogen, Paisley, UK; catalogue number 37-2900) and IRDye® 800CW Goat (polyclonal) Anti-Mouse IgG (H + L), Highly Cross Adsorbed secondary antibody (LI-COR Biosciences, Lincoln, USA; catalogue number 926-32210) at 1:500 and 1:5000 dilutions, respectively. The solubilization yields for each detergent were determined by calculating the % of Etx pore in the supernatant relative to its amount in the total protein sample using Image Studio ver 5.2 software (LI-COR Biosciences, Lincoln, USA).

**Epsilon pore purification**. For DDM-extracted samples, solubilized Etx oligomers were bound to Ni-NTA resin (Qiagen), washed with 20 volumes of DPBS containing 20 mM imidazole and 0.02% (w/v) DDM, transferred to a Bio-Rad gravity-flow column, washed again with 20 volumes of DPBS containing 20 mM imidazole and 0.02% (w/v) DDM, and eluted in three volumes of DPBS containing 200 mM imidazole and 0.02% (w/v) DDM.

**Binding and oligomerization**. Super Dome cells were grown to confluence in 6-well plates ($1.2 \times 10^6$ cells per well), and after washing cells three times with DPBS to remove residual trypsin, each well was incubated with 2 ml pre-warmed DMEM-F12 medium containing trypsin-activated recombinant Etx wild-type, Etx-D250A or Etx-H162A (391 ng ml$^{-1}$, equivalent to 5× CT$_{50}$ dose of wild-type Etx) at 37 °C for 1 h to allow binding and/or oligomerization. Cells were subsequently washed three times with DPBS to remove unbound Etx, lysed by three cycles of freezing and thawing in 100 μl DPBS supplemented with Halt™ Protease Inhibitor Cocktail (100×) (Thermo Fisher Scientific, UK) at a final concentration of 1× and harvested by scraping. The collected cells were transferred to a microcentrifuge tube and solubilized in DPBS containing 1% (w/v) DDM by incubation at 4 °C overnight with occasional shaking. After incubation, insoluble material was removed by centrifugation at $20,000 \times g$ for 30 min at 4 °C and the concentration of solubilized protein was determined by BCA assay.

Toxin complexes or monomers in solubilization buffer were analysed by native-PAGE and SDS-PAGE. Solubilized proteins were resolved by NuPAGE™ 7% Tris-Acetate Protein Gels (Invitrogen Ltd., Paisley, UK) using Surelock Xcell apparatus (Invitrogen Ltd, Paisley, UK). For protein gel electrophoresis under denaturing running conditions NuPAGE™ Tris-Acetate SDS Running Buffer (Invitrogen Ltd, Paisley, UK) and NuPAGE LDS sample buffer (Invitrogen Ltd., Paisley, UK) were used. For protein gel electrophoresis under native running conditions Novex™ Tris-Glycine Native Running Buffer (Invitrogen Ltd., Paisley, UK) and Novex™ Tris-Glycine Native Sample Buffer (Invitrogen Ltd., Paisley, UK) were used. Denaturing gels were electrophoresed at 150 V for 1 h at room temperature, while native gels were electrophoresed at 150 V for 2 h in ice cold running buffer. For SDS-PAGE the molecular weight standard Perfect Protein™ Marker, 10–225 kDa (Merck, Darmstadt, Germany) was used. For native-PAGE, the molecular weight marker NativeMark™ Unstained Protein Standard (Invitrogen Ltd., Paisley, UK) was used.

After electrophoresis, proteins in the gel were transferred to a nitrocellulose membrane using the Trans-Blot® Turbo™ Transfer System (Bio-Rad). The membrane was blocked with PBS containing 0.1% Tween-20 and 5% skim milk and bound P-Etx and/or oligomeric complexes were detected with rabbit anti-Etx polyclonal primary antibody[51] at 1:500 dilution and IRDye® 800CW goat anti-rabbit IgG (H + L) secondary antibody (LI-COR Biosciences, Lincoln, USA; catalogue number 926-32211) at 1:5000 dilution. For loading control, the mouse monoclonal Na$^+$/K$^+$-ATPase α1 (C464.6): sc-21712 primary antibody (Santa Cruz Biotechnology; catalogue number sc-21712) and the IRDye® 680RD goat (polyclonal) anti-mouse IgG (H + L) secondary antibody (LI-COR Biosciences, Lincoln, USA; catalogue number 925-68070) were used at 1:200 and 1:5000 dilutions, respectively. Antibodies were diluted in PBS containing 0.1% Tween-20 and 3% skim milk. Immuno-reactive bands were detected using the Odyssey CLx infrared imaging system (LI-COR Biosciences, Lincoln, USA). Quantitative analyses were performed using the Image Studio ver 5.2 software and the Housekeeping Protein Normalization Protocol (LI-COR Biosciences, Lincoln, USA). The oligomer yield for each Etx variant was determined by calculating the percentage of fluorescence intensity relative to wild-type toxin.

**Negative stain EM**. Purified recombinant P-Etx variants were trypsin activated as described above. After activation, samples were diluted to 0.02 mg ml$^{-1}$ and 3 μl was added to freshly glow-discharged carbon-coated grids for 60 s before staining with 20 μl of 2% (w/v) aqueous uranyl acetate. Grids were then imaged on a Philips CM200 FEG transmission electron microscope operating at an accelerating voltage of 200 kV. Images were recorded at a calibrated pixel size of 2 Å on a TVIPS F224HD CCD camera.

**Specimen preparation and data collection**. Etx eluate fraction 4 (~0.04 mg ml$^{-1}$) was diluted 3-fold in elution buffer (DPBS containing 20 mM imidazole and 0.02% (w/v) DDM) and deposited on Quantifoil R1.2/1.3 grids overlaid with graphene oxide, as described in ref. [20]. Grids were then blotted and plunge frozen using a custom-made device and stored in LN2 until further use. Data were collected on an FEI Titan Krios operating at an accelerating voltage of 300 kV. All datasets were collected on an FEI Falcon 3EC direct electron detector at a nominal magnification

of ×75K (calibrated pixel size of 1.07 Å) and a dose rate of 0.6e− per pixel per second in counting mode over 60 s and 75 fractions. Conventionally defocused and phase plate datasets were collected using the EPU software (Thermo Fisher Scientific). In brief, grid squares and suitable holes were selected and microscope alignments were performed. Just prior to data collection, the phase plate was inserted and the un-scattered beam was made parallel by observing the Ronchigram in the back focal plane of the objective lens over an empty specimen area. Condenser astigmatism was corrected by reducing the smearing of the Ronchigram near the blow-up point. Due to the extremely low screen current used (~0.014 nA), the 150 μm C2 aperture was used to aid in this process and then switched back to the 50 μm aperture. Coma and astigmatism were corrected using AutoCTF. Defocused data were collected by defocusing between −2 and −3.5 μm every 15 μm, and phase plate data between −0.3 and −0.8 μm with focusing at every hole.

**Image processing**. All datasets were processed in Relion-3[52]. Micrographs were first corrected for large movements using MotionCorr2[53] and a 4 × 4 patch with no grouping. CTF parameters were estimated using GTCF[54]. Autopicking was performed in Relion after creating references from manually picked particles. The processing procedure is summarized in Supplementary Fig. 4. The resolution was calculated using Gold-Standard 0.143 criterion, resulting in a 4.6 Å map for the defocused dataset and 3.2 Å for the phase plate dataset. The variation in resolution in the map was calculated using the Relion local resolution implementation. Box sizes of 220 pixels were used for both the defocused and phase plate datasets. To ensure that high-resolution information was not lost by the high defocus values used for the former dataset, particles were re-extracted in 400 pixel boxes and subjected to 3D refinement. No increase in resolution was observed for the larger box size particles. B-factor plots were calculated using the bfactor_plot.py scripts provided with Relion.

**Model building and refinement**. A model was built into the 3.2 Å map by initially docking the RBD of wild-type Etx crystal structure (PDB: 1UYJ) and then extending this using Coot. Refinement of the atomic coordinates was performed using Phenix[55] real space refine and re-iterating the model building-refinement procedure several times.

**Statistics**. Data were analysed using the Prism v7 software (GraphPad Software Inc., La Jolla, CA, USA). Data are expressed as the mean value ± SEM.

**Reporting summary**. Further information on research design is available in the Nature Research Reporting Summary linked to this article.

## Data availability
Data supporting the findings of this manuscript are available from the corresponding author upon reasonable request. A reporting summary for this Article is available as a Supplementary Information file.

The cryo-EM maps of Etx pore have been deposited to the Electron Microscopy Data Bank under accession number EMD-4789 and the refined atomic coordinates have been deposited to the Protein Data Bank under accession number 6RB9. The source data for Fig. 1 and Supplementary Figs. 2, 12–15 are provided as a Source Data file.

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

## Acknowledgements

We thank Professor Nicholas Harmer and Alice Cross, University of Exeter, UK for assistance with the thermostability assay. We also wish to thank Dr. Shaoxia Chen, Dr. Giuseppe Cannone, Dr. Greg McMullan, MRC Laboratory of Molecular Biology, Cambridge, UK; Dr. Peter Moody, University of Leicester, UK; and Dr. Edmund Kunji, MRC Mitochondrial Biology Unit, Cambridge, UK for helpful discussions and advice. This work was supported by grants from the UK Medical Research Council (MC-A021-53019) and the Wellcome Trust (WT089618MA).

## Author contributions

C.G.S. collected the cryo-EM data, performed the image processing and model building, analysed the data and wrote the paper. A.R.C. contributed to model building, analysed the data and wrote the paper. C.E.N., M.R.P., D.S.M., A.K.B. and R.W.T. analysed the data and wrote the paper. M.B.-B. purified proteins, assembled and purified the pore complex, conducted the cytotoxicity, thermostability and oligomerization assays, generated D250A site-directed mutant, developed experimental methods, analysed the data, wrote the paper and coordinated the study. D.S.M., A.K.B. and R.W.T. initiated the project.

## Additional information

**Competing interests:** The authors declare no competing interests.

