## [Peer Review File · Nature Communications]

Reviewers' Comments:

Reviewer #1:

Remarks to the Author:

In this manuscript, Savva et al. determined cryo EM structure of Etx pore. According to the structure revealed, they constructed D250A mutant and analyzed its activity. Further, combining with previous studies, they proposed important interactions and importance of double beta-barrel structure. Etx is the third strongest toxin and deeply studied, and its pore structure has been desired to reveal its molecular mechanism for a long period. Therefore, this study which determined its atomic resolution structure has a great importance, and this reviewer felt that this study has a potential to be published in Nat. Commun.

However, in the present form, the authors spend too much for Introduction section and explanation of sample preparation and structure determination in both Results and Discussion sections, and consequently, do not discuss deeply about dynamic molecular mechanism of Etx which is the most important point of this study. Therefore, this reviewer asks the authors to revise the manuscript thoroughly so that what this new pore structure revealed is discussed deeply combining with previous studies accumulated. Since they know both monomer and pore structures in atomic resolution, mechanism of assembly and conformation change including driving force for the large structure change should be described in particular.

From descriptions from line 313, the authors propose importance of the double-barrel structure for conformation change. However, there is no evidence to support it, and the propose is too speculative. Authors should logically explain the mechanism, with referring previous experimental evidence to support it. If necessary, they may have to carry out biochemical experiments to reveal the mechanism.

As well, residues proposed to be essential is limited. Much comprehensive discussion should be held, using the atomic resolution pore structure.

Although the present manuscript should be revised thoroughly, there is no doubt that this cryo-EM structure is a major progress in Etx research.

Reviewer #2:

Remarks to the Author:

Review of NCOMMS-18-38388

Corresponding Author: Monika Bokori-Brown

Comments for the Author

When composing your report, the following questions might assist you in writing an incisive, well-justified review. What are the major claims of the paper? Are they novel and will they be of interest to others in the community and the wider field? If the conclusions are not original, it would be helpful if you could provide relevant references. Is the work convincing, and if not, what further evidence would be required to strengthen the conclusions? On a more subjective note, do you feel that the paper will influence thinking in the field? Please feel free to raise any further questions and concerns about the paper.

We would also be grateful if you could comment on the appropriateness and validity of any statistical analysis, as well the ability of a researcher to reproduce the work, given the level of detail provided.

To increase the transparency and openness of the reviewing process, we do support our reviewers signing their reports to authors if the reviewers feel comfortable doing so. If, however, you prefer

to send an anonymized report we will continue to respect and maintain your anonymity. Referee reports, whether signed or not, are subsequently shared with the other reviewers.

In their manuscript Savva et al. present the 3.2 Å structure of *Clostridium perfringens* epsilon toxin (Etx) pore, a member of the aerolysin pore forming toxin family. Like lysenin and aerolysin pores, for which a structure has been reported, Etx pore is heptameric, it has a beta-barrel spanning the hole length of the protein, and the extracellular end of this barrel is surrounded by a second beta-barrel, termed cap domain. The newly reported structure supports the hypothesis that a double-beta barrel is a hallmark of aerolysin-family pores. This feature was previously proposed to play an important role for the high stability of the pores. The results presented here suggest that, in addition to this function, the establishment of a double barrel is necessary for pore formation. The structure explains the function of a number of amino-acids involved in receptor binding, oligomer and pore formation, which were previously highlighted by mutant phenotypes. Purifying Etx pores, for which no cellular receptor has yet been identified, required solubilizing pores assembled at the surface of culture cells. Several detergent had to be assessed. This approach is both original and challenging in the context of structural studies. A further challenge followed the small size of Etx pore. The authors remarkably managed to improve Etx pore map resolution from 4.6 Å to 3.2 Å by using a volta phase plate.

The results are novel and of high relevance for the field of pore forming proteins, as well as for the protein folding community. The resolution is sufficient to confidently model the position of side chains, which brings significant information about the interactions leading to pore formation and stability. Even though the receptor(s) for Etx is not yet known, receptor binding residues have been identified in Etx. The structure presented here shows their localisation and suggest that Etx binds its receptor very close to the membrane bilayer.

The structure shows that H162 located in the inner beta-barrel and D250 situated in the cap domain form a hydrogen bond. In order to assess the role of this bond, the authors have mutated D250 to an alanine. They observed that the mutant was not cytotoxic, i.e. it did not form pore. Their thermal stability analysis suggests that the mutant monomer is correctly folded. It is not clear however if this mutant does not form heptameric prepore or if the prepore cannot undergo the conformational changes necessary to form a pore. The authors mentioned that they could not detect soluble prepores in the H162A mutant by negative stain EM analysis. Was D250A analysed by negative stain EM? It would be interesting to see an SDS-PAGE, as well as a native-PAGE of the mutants vs. the wild type after protease cleavage activation. This should reveal whether heptamers are formed at all and how stable they are.

Minor points:

II. 207-208: It would be helpful to show in a figure the salt bridges that are listed here.

II. 214-216: The paper that first described aerolysin rivet should be cited :

10.1038/sj.emboj.7600959 Iacovache et al. EMBO J 2006.

II. 305-307. Reference 21 does not report that Etx Y84 and F105 are important for Etx function. The author should fix this and cite the correct reference.

Sup. Fig. 1. What is the band slightly below 75 kDa?

Sup. Fig. 2. The authors could box a few particles as they did in Fig. 2.

Sup. Fig. 5 legend : should read « was used to collect» and not « was used top collect »

Provided that the points that I raised above are addressed I strongly recommend publication.

Reviewer #1 (Remarks to the Author):

In this manuscript, Savva et al. determined cryo EM structure of Etx pore. According to the structure revealed, they constructed D250A mutant and analyzed its activity. Further, combining with previous studies, they proposed important interactions and importance of double beta-barrel structure. Etx is the third strongest toxin and deeply studied, and its pore structure has been desired to reveal its molecular mechanism for a long period. Therefore, this study which determined its atomic resolution structure has a great importance, and this reviewer felt that this study has a potential to be published in Nat. Commun.

However, in the present form, the authors spend too much for Introduction section and explanation of sample preparation and structure determination in both Results and Discussion sections, and consequently, do not discuss deeply about dynamic molecular mechanism of Etx which is the most important point of this study. Therefore, this reviewer asks the authors to revise the manuscript thoroughly so that what this new pore structure revealed is discussed deeply combining with previous studies accumulated. Since they know both monomer and pore structures in atomic resolution, mechanism of assembly and conformation change including driving force for the large structure change should be described in particular.

From descriptions from line 313, the authors propose importance of the double-barrel structure for conformation change. However, there is no evidence to support it, and the propose is too speculative. Authors should logically explain the mechanism, with referring previous experimental evidence to support it. If necessary, they may have to carry out biochemical experiments to reveal the mechanism.

As well, residues proposed to be essential is limited. Much comprehensive discussion should be held, using the atomic resolution pore structure.

Although the present manuscript should be revised thoroughly, there is no doubt that this cryo-EM structure is a major progress in Etx research.

Point-by-point response to Reviewer #1's comments

Reviewer: 'the authors spend too much for Introduction section and explanation of sample preparation and structure determination in both Results and Discussion sections, and consequently, do not discuss deeply about dynamic molecular mechanism of Etx which is the most important point of this study.'

Response: We are grateful for the insightful comments of the reviewer. We agree with the reviewer that the Introduction section was too long and have shortened it to less than 1,000 word, which also complies with the editorial policies of the publisher.

We believe that explanation of the structural determination steps is an important part of this study. The formation of the pore complex on target cells and the use of the phase plate for this sub-optimal sample were critical for the structural determination of the complex and could be of benefit to others.

Unlike aerolysin, Etx does not spontaneously oligomerize into SDS-resistant species when activated in aqueous buffer, and negative stain electron microscopy experiments we have performed since (see below) did not reveal regular particles other than elongated monomers and small clumps. Furthermore, efficient oligomerization cannot be promoted by the addition of liposomes of various lipid compositions or detergent micelles. Therefore, the only viable way to study the pore form of Etx was to isolate the complexes directly from cultured cells known to be susceptible to Etx. Purifying Etx pores, for which no cellular receptor has yet been identified, required solubilizing pores assembled at the surface of culture cells. Several detergents had to be assessed, an approach that is both original and challenging in the context of structural studies. A further challenge was the heterogeneity of the purified pore sample, as the extremely low yields meant we could not pursue additional clean-up steps, such as size-exclusion chromatography. Remarkably, this obstacle has been overcome by use of the Volta phase plate that improved Etx pore map resolution from 4.6 Å to 3.2 Å. As both sample preparation and structure determination were highly challenging tasks and critical to obtaining the pore structure of Etx, we strongly believe that their detailed explanation in both the Results and

Discussion sections is justified. We have revised the Results section to further highlight the challenges of sample preparation (lines 120-125):

‘The formation of biologically active oligomers has been a limiting factor for obtaining structural information of the pore form of Etx. Unlike aerolysin^{37, 38}, Etx does not spontaneously oligomerize into SDS-resistant species when activated in aqueous buffer, and negatively stained EM samples do not reveal regular particles other than elongated monomers and small clumps (Supplementary Fig. 1a). Furthermore, efficient oligomerization cannot be promoted by the addition of liposomes of various lipid compositions or detergent micelles.’

We have also revised the Supplementary Information file to include:

‘Supplementary Fig. 1. Negative stain electron microscopy of Etx variants.’

We agree with the reviewer that the dynamic molecular mechanism of Etx was not discussed in enough detail. We have revised the manuscript thoroughly so that what this new pore structure revealed is discussed deeply, combining with previous studies accumulated.

To investigate the importance of the H162-D250 interaction we performed additional experiments (binding and oligomerisation assays; Supplementary Figures 12-14) and have revised the Results section accordingly (lines 245-280):

‘To investigate the importance of the H162-D250 interaction, we constructed D250A and H162A mutants. Etx-H162A (corresponding to H149A in⁴⁵) has previously been reported to have reduced, but not abolished, toxicity⁴⁵, and the crystal structure of P-Etx-H162A closely resembles that of wild type P-Etx²², while purified recombinant Etx-D250A is completely inactive towards Super Dome cells, as shown in Supplementary Figure 11a, and thermal stability assays indicate that P-Etx-D250A behaves similarly to wild type P-Etx (Supplementary Figure 11b), suggesting that P-Etx-D250A is correctly folded. Examination of D250 and H162A in the monomeric Etx crystal structure indicates that D250A and H162 are surface exposed residues and are not involved in any interactions, suggesting that the importance of these residues lies in the H162-D250 hydrogen bond formed in the oligomer. SDS-PAGE analysis of oligomer formation revealed that both Etx-D250A and Etx-H162A produce SDS-resistant oligomers of the same apparent molecular weight as wild type Etx (approximately 155 kDa, consistent with previous reports^{2, 28, 29, 34}). When Super Dome cells were incubated with activated wild type Etx, Etx-D250A or Etx-H162A (Supplementary Figure 12a, lanes 11-13), oligomers were formed with a mean oligomer yield of 5.09 ± 2.62 % (S.E.M.) for Etx-D250A relative to wild type Etx (Supplementary Figure 12b), albeit in a concentration dependent manner (Supplementary Figure 13), and with a mean oligomer yield of 44.54 ± 10.25 % (S.E.M.) for Etx-H162A (Supplementary Figure 12b). These results suggest that while D250A oligomers resembling the wild type ones, either in a pre-pore or post pre-pore state, can assemble, they cannot produce pores that have a cytotoxic effect on cells (Supplementary Figure 11a), even at the highest concentration of 6 μ M tested (2,426 times the CT₅₀ dose of wild type Etx). Further analysis of oligomer formation by native-PAGE revealed a much larger complex of approximately 624 kDa relative to the SDS-resistant oligomers of approximately 155 kDa when Super Dome cells were incubated with activated wild type Etx (Supplementary Figure 14, lane 10). The same complex, albeit at lower intensity, was also observed when cells were incubated with activated Etx-H162A (Supplementary Figure 14, lane 12), but was not detectable when cells were incubated with activated Etx-D250A (Supplementary Figure 14, lane 11), providing further evidence that D250A oligomers detected by SDS-PAGE may not correspond to pores. Both D250A and H162A mutants retain their ability to bind to Super Dome cells (Supplementary Figure 12a, lanes 8-10, and Supplementary Figure 14a, lanes 7-9), suggesting that residues D250 and H162 are important for events downstream of receptor binding. Similar to activated wild type Etx, negative stain EM did not reveal regular particles other than elongated monomers and small clumps for activated Etx-D250A and Etx-H162A (Supplementary Figure 1).’

We have also revised the Materials section to include a detailed description of the additional ‘Binding and oligomerisation’ and ‘Negative stain electron microscopy’ experiments we have carried out (lines 523-576). In addition, we have revised the Supplementary Information file to include:

Supplementary Figure 12. SDS-PAGE analysis of oligomer formation

Supplementary Figure 13. Etx-D250A forms oligomers in a concentration-dependent manner
Supplementary Figure 14. Native-PAGE analysis of oligomer formation.

We have also made significant revisions to the Discussion section to provide a detailed discussion of the dynamic molecular mechanisms revealed by the Etx pore structure, with previous studies accumulated (lines 323-359):

‘A number of interesting mutants of Etx have been studied in the past whilst trying to understand how Etx transitions from the water-soluble to the pore form. Mutating H149 (H162 in our model) to alanine resulted in a significant decrease in activity of the recombinant toxin both *in vitro* and *in vivo*⁴⁵. However, substitution with serine at position 162 does not abolish lethal activity⁴⁵, suggesting that H162A is less active not because of removal of the bulky imidazole side chain but possibly due to the loss of a hydrogen bond donor at this position. H162, located in the membrane-distal part of the inner β -barrel, has previously been equated to aerolysin Y221, and recent studies proposed that, similar to the aerolysin Y221G mutant, the Etx H162A mutant may prevent the β -hairpin domain involved in pore formation from unfolding^{19, 41}. However, unlike the aerolysin Y221G mutant, the Etx H162A mutant did not reveal water-soluble, heptameric pre-pore particles by negative stain EM (Supplementary Figure 1b).

Examination of the Etx pore structure reveals that H162 may also play a role in stabilizing the DBB fold by forming a hydrogen bond with D250 in the cap domain. This interaction will stabilize the DBB fold further in addition to the hydrophobic interactions and make it more rigid. In agreement with this hypothesis, substitution of aspartate with alanine at position 250 results in a completely inactive Etx that displays approximately 95% reduction in SDS-resistant oligomers relative to wild type Etx. The reduced oligomerization of Etx-D250A and its complete inactivity towards Super Dome cells could be a result of either oligomer instability or a reduction in oligomer formation. Since D250 does not participate in any inter-chain interactions in the oligomer we cannot explain the latter. The abolished cytotoxicity of Etx-D250A suggests that the SDS-resistant oligomers resembling the wild type ones may correspond to pre-pores or post pre-pores. This may explain why Etx-D250A oligomers resembling the wild type ones were not observed by native-PAGE immunoblot. Pre-pores or post pre-pores are expected to migrate differently to pores on native-PAGE as they will lack any associated detergents or lipids.

Further evidence for the importance of the DBB stability in pore formation also comes from previous mutagenesis studies. Tyrosine 84 in the cap domain and F105 in the inner β -barrel have previously been identified as important to Etx function⁴⁷. Y84A completely abolishes pore formation and is only recoverable by substitution to another aromatic residue, whilst F105A has reduced cytotoxic activity towards MDCK cells at low toxin concentrations⁴⁷. In the pore structure these two residues form a π - π stacking interaction (Figure 3d and Supplementary Figure 8b). Similar to D250A and H162A mutants, Y84A and F105A mutants also retain their ability to bind to MDCK cells.’

Reviewer: ‘Since they know both monomer and pore structures in atomic resolution, mechanism of assembly and conformation change including driving force for the large structure change should be described in particular.’

Response: To illustrate the structural rearrangements during pore formation we have constructed a movie (**Supplementary Movie 1**). In addition, we have constructed a hypothetical pre-pore to describe the mechanism of assembly and the transition of the soluble monomers to the membrane-inserted pore form (**Supplementary Fig. 15**), and have revised the results section with the following paragraph (lines 291-305):

‘To further understand how Etx assembles into an oligomer and the rearrangements required for transition to the pore state, we constructed a hypothetical pre-pore by superimposing the crystal structure monomer to the RBD of the pore structure and imposing 7-fold symmetry, similar to hypothetical pre-pores constructed previously for lysenin^{20, 35}. This allows us to observe some of the displacements that have to occur prior to pre-pore formation (Supplementary Figure 15). Calculations of molecular clashes (inter-atom overlap >0.6 Å) highlight the potential regions that would have to rearrange upon oligomerisation. It is clear from these clashes that the CTP, as expected, severely

obstructs oligomerization, and the pre-pore model illustrates why this region must be removed prior to monomer association. In addition to the CTP, other significant clashes occur near the oligomerisation surfaces on either side of a monomer, indicating again that some rearrangement will have to happen. Finally, the hypothetical insertion loop or “tongue” also creates clashes upon monomer association and, as described previously for lysenin³⁵, the displacement of the tongue region may be the driving force for the pre-insertion strands to unfold and the formation of the inner β -barrel.’

Reviewer: ‘From descriptions from line 313, the authors propose importance of the double-barrel structure for conformation change. However, there is no evidence to support it, and the propose is too speculative. Authors should logically explain the mechanism, with referring previous experimental evidence to support it. If necessary, they may have to carry out biochemical experiments to reveal the mechanism.

Response: As we mentioned above, we have carried out additional biochemical experiments (negative stain electron microscopy, oligomerisation and binding assays) that support the importance of the double β -barrel structure in pore formation. We have made significant revisions to the Results and Discussion sections to explain the mechanism, with previous studies accumulated (please see above).

Reviewer: ‘residues proposed to be essential is limited. Much comprehensive discussion should be held, using the atomic resolution pore structure.’

Response: As we mentioned above, we have made significant revisions to the Discussion section to provide a detailed discussion of the dynamic molecular mechanisms revealed by the Etx pore structure, with previous studies accumulated (lines 323-359). The pore structure also explains the function of a number of amino-acids involved in receptor binding, oligomer and pore formation, which were previously highlighted by mutant phenotypes. The pore structure of Etx shows that the receptor binding residues that have previously been identified are located directly above the membrane in a position that would facilitate interaction with a receptor.

Reviewer #2 (Remarks to the Author):

In their manuscript Savva *et al.* present the 3.2 Å structure of *Clostridium perfringens* epsilon toxin (Etx) pore, a member of the aerolysin pore forming toxin family. Like lysenin and aerolysin pores, for which a structure has been reported, Etx pore is heptameric, it has a beta-barrel spanning the hole length of the protein, and the extracellular end of this barrel is surrounded by a second beta-barrel, termed cap domain. The newly reported structure supports the hypothesis that a double-beta barrel is a hallmark of aerolysin-family pores. This feature was previously proposed to play an important role for the high stability of the pores. The results presented here suggest that, in addition to this function, the establishment of a double barrel is necessary for pore formation. The structure explains the function of a number of amino-acids involved in receptor binding, oligomer and pore formation, which were previously highlighted by mutant phenotypes.

Purifying Etx pores, for which no cellular receptor has yet been identified, required solubilizing pores assembled at the surface of culture cells. Several detergent had to be assessed. This approach is both original and challenging in the context of structural studies. A further challenge followed the small size of Etx pore. The authors remarkably managed to improve Etx pore map resolution from 4.6 Å to 3.2 Å by using a volta phase plate.

The results are novel and of high relevance for the field of pore forming proteins, as well as for the protein folding community. The resolution is sufficient to confidently model the position of side chains, which brings significant information about the interactions leading to pore formation and stability. Even though the receptor(s) for Etx is not yet known, receptor binding residues have been identified in Etx. The structure presented here shows there localisation and suggest that Etx binds its receptor very close to the membrane bilayer.

The structure shows that H162 located in the inner beta-barrel and D250 situated in the cap domain form a hydrogen bond. In order to assess the role of this bond, the authors have mutated D250 to an alanine. They observed that the mutant was not cytotoxic, i.e. it did not form pore. Their thermal stability analysis suggests that the mutant monomer is correctly folded. It is not clear however if this

mutant does not form heptameric prepore or if the prepore cannot undergo the conformational changes necessary to form a pore. The authors mentioned that they could not detect soluble prepores in the H162A mutant by negative stain EM analysis. Was D250A analysed by negative stain EM? It would be interesting to see an SDS-PAGE, as well as a native-PAGE of the mutants vs. the wild type after protease cleavage activation. This should reveal whether heptamers are formed at all and how stable they are.

Minor points:

ll. 207-208: It would be helpful to show in a figure the salt bridges that are listed here.

ll. 214-216: The paper that first described aerolysin rivet should be cited : 10.1038/sj.emboj.7600959 Iacovache et al. EMBO J 2006.

ll. 305-307. Reference 21 does not report that Etx Y84 and F105 are important for Etx function. The author should fix this and cite the correct reference.

Sup. Fig. 1. What is the band slightly below 75 kDa?

Sup. Fig. 2. The authors could box a few particles as they did in Fig. 2.

Sup. Fig. 5 legend : should read « was used to collect» and not « was used top collect »

Point-by-point response to Reviewer #2's comments

Reviewer: 'The structure shows that H162 located in the inner beta-barrel and D250 situated in the cap domain form a hydrogen bond. In order to assess the role of this bond, the authors have mutated D250 to an alanine. They observed that the mutant was not cytotoxic, i.e. it did not form pore. Their thermal stability analysis suggests that the mutant monomer is correctly folded. It is not clear however if this mutant does not form heptameric prepore or if the prepore cannot undergo the conformational changes necessary to form a pore.'

Response: We are grateful for the insightful comments of the reviewer. To investigate the importance of the H162-D250 interaction we performed additional experiments (binding and oligomerisation assays; Supplementary Figures 12-14) and have revised the Results section accordingly (lines 245-280):

'To investigate the importance of the H162-D250 interaction, we constructed D250A and H162A mutants. Etx-H162A (corresponding to H149A in⁴⁵) has previously been reported to have reduced, but not abolished, toxicity⁴⁵, and the crystal structure of P-Etx-H162A closely resembles that of wild type P-Etx²², while purified recombinant Etx-D250A is completely inactive towards Super Dome cells, as shown in Supplementary Figure 11a, and thermal stability assays indicate that P-Etx-D250A behaves similarly to wild type P-Etx (Supplementary Figure 11b), suggesting that P-Etx-D250A is correctly folded. Examination of D250 and H162A in the monomeric Etx crystal structure indicates that D250A and H162 are surface exposed residues and are not involved in any interactions, suggesting that the importance of these residues lies in the H162-D250 hydrogen bond formed in the oligomer. SDS-PAGE analysis of oligomer formation revealed that both Etx-D250A and Etx-H162A produce SDS-resistant oligomers of the same apparent molecular weight as wild type Etx (approximately 155 kDa, consistent with previous reports^{2, 28, 29, 34}). When Super Dome cells were incubated with activated wild type Etx, Etx-D250A or Etx-H162A (Supplementary Figure 12a, lanes 11-13), oligomers were formed with a mean oligomer yield of 5.09 ± 2.62 % (S.E.M.) for Etx-D250A relative to wild type Etx (Supplementary Figure 12b), albeit in a concentration dependent manner (Supplementary Figure 13), and with a mean oligomer yield of 44.54 ± 10.25 % (S.E.M.) for Etx-H162A (Supplementary Figure 12b). These results suggest that while D250A oligomers resembling the wild type ones, either in a pre-pore or post pre-pore state, can assemble, they cannot produce pores that have a cytotoxic effect on cells (Supplementary Figure 11a), even at the highest concentration of 6 μ M tested (2,426 times the CT₅₀ dose of wild type Etx). Further analysis of oligomer formation by native-PAGE revealed a much larger complex of approximately 624 kDa relative to the SDS-resistant oligomers of approximately 155 kDa when Super Dome cells were incubated with activated wild type Etx (Supplementary Figure 14, lane 10). The same complex, albeit at lower intensity, was also observed when cells were incubated with activated Etx-H162A (Supplementary Figure 14, lane 12), but was not detectable when cells were incubated with activated Etx-D250A (Supplementary Figure 14, lane 11), providing further evidence that D250A oligomers detected by SDS-PAGE may not correspond to pores. Both D250A and H162A mutants retain their ability to bind to Super Dome cells

(Supplementary Figure 12a, lanes 8-10, and Supplementary Figure 14a, lanes 7-9), suggesting that residues D250 and H162 are important for events downstream of receptor binding. Similar to activated wild type Etx, negative stain EM did not reveal regular particles other than elongated monomers and small clumps for activated Etx-D250A and Etx-H162A (Supplementary Figure 1).’

We have also revised the Materials section to include a detailed description of the additional ‘Binding and oligomerisation’ and ‘Negative stain electron microscopy’ experiments we have carried out (lines 523-576). In addition, we have revised the Supplementary Information file to include:

Supplementary Figure 12. SDS-PAGE analysis of oligomer formation

Supplementary Figure 13. Etx-D250A forms oligomers in a concentration-dependent manner.

Supplementary Figure 14. Native-PAGE analysis of oligomer formation.

To highlight the importance of the double β -barrel structure in pore formation, we have made significant revisions to the Discussion section (lines 323-359):

‘A number of interesting mutants of Etx have been studied in the past whilst trying to understand how Etx transitions from the water-soluble to the pore form. Mutating H149 (H162 in our model) to alanine resulted in a significant decrease in activity of the recombinant toxin both *in vitro* and *in vivo*⁴⁵. However, substitution with serine at position 162 does not abolish lethal activity⁴⁵, suggesting that H162A is less active not because of removal of the bulky imidazole side chain but possibly due to the loss of a hydrogen bond donor at this position. H162, located in the membrane-distal part of the inner β -barrel, has previously been equated to aerolysin Y221, and recent studies proposed that, similar to the aerolysin Y221G mutant, the Etx H162A mutant may prevent the β -hairpin domain involved in pore formation from unfolding^{19,41}. However, unlike the aerolysin Y221G mutant, the Etx H162A mutant did not reveal water-soluble, heptameric pre-pore particles by negative stain EM (Supplementary Figure 1b).

Examination of the Etx pore structure reveals that H162 may also play a role in stabilizing the DBB fold by forming a hydrogen bond with D250 in the cap domain. This interaction will stabilize the DBB fold further in addition to the hydrophobic interactions and make it more rigid. In agreement with this hypothesis, substitution of aspartate with alanine at position 250 results in a completely inactive Etx that displays approximately 95% reduction in SDS-resistant oligomers relative to wild type Etx. The reduced oligomerization of Etx-D250A and its complete inactivity towards Super Dome cells could be a result of either oligomer instability or a reduction in oligomer formation. Since D250 does not participate in any inter-chain interactions in the oligomer we cannot explain the latter. The abolished cytotoxicity of Etx-D250A suggests that the SDS-resistant oligomers resembling the wild type ones may correspond to pre-pores or post pre-pores. This may explain why Etx-D250A oligomers resembling the wild type ones were not observed by native-PAGE immunoblot. Pre-pores or post pre-pores are expected to migrate differently to pores on native-PAGE as they will lack any associated detergents or lipids.

Further evidence for the importance of the DBB stability in pore formation also comes from previous mutagenesis studies. Tyrosine 84 in the cap domain and F105 in the inner β -barrel have previously been identified as important to Etx function⁴⁷. Y84A completely abolishes pore formation and is only recoverable by substitution to another aromatic residue, whilst F105A has reduced cytotoxic activity towards MDCK cells at low toxin concentrations⁴⁷. In the pore structure these two residues form a π - π stacking interaction (Figure 3d and Supplementary Figure 8b). Similar to D250A and H162A mutants, Y84A and F105A mutants also retain their ability to bind to MDCK cells.’

Reviewer: ‘The authors mentioned that they could not detect soluble prepores in the H162A mutant by negative stain EM analysis. Was D250A analysed by negative stain EM?’

Response: Unlike aerolysin, Etx does not spontaneously oligomerize into SDS-resistant species when activated in aqueous buffer, and additional negative stain electron microscopy experiments we have carried out on trypsin activated Etx-D250A alongside trypsin activated Etx wild type and Etx-H162A did not reveal regular particles other than elongated monomers and small clumps.

We have revised the Results section to include these data (lines 120-125):

‘The formation of biologically active oligomers has been a limiting factor for obtaining structural information of the pore form of Etx. Unlike aerolysin^{37,38}, Etx does not spontaneously oligomerize into SDS-resistant species when activated in aqueous buffer, and negatively stained EM samples do not reveal regular particles other than elongated monomers and small clumps (Supplementary Fig. 1a). Furthermore, efficient oligomerization cannot be promoted by the addition of liposomes of various lipid compositions or detergent micelles.’

We have also revised the Supplementary Information file to include ‘**Supplementary Fig. 1. Negative stain electron microscopy of Etx variants.**’, and the Methods section to include ‘Negative stain electron microscopy’ (lines 570-576).

Reviewer: ‘It would be interesting to see an SDS-PAGE, as well as a native-PAGE of the mutants vs. the wild type after protease cleavage activation. This should reveal whether heptamers are formed at all and how stable they are.’

Response: As we mentioned above, unlike aerolysin, Etx does not spontaneously oligomerize into SDS-resistant species when activated in aqueous buffer. To investigate the importance of the H162-D250 interaction we performed additional experiments by incubating Super Dome cells with trypsin-activated Etx (wild type, H162A or D250A) alongside buffer only control at 37 °C for 1 h to allow binding and oligomerisation. Following detergent solubilisation (1% (w/v) DDM), toxin complexes were separated using SDS-PAGE and native-PAGE and oligomerization was assessed by immunoblotting with anti-Etx polyclonal antibody (**Supplementary Figures 12-14**). As we mentioned above, we have revised the Results section to include SDS-PAGE and native-PAGE analysis of oligomer formation (lines 255-274), and associated Materials and Discussion sections (please see above).

Minor points:

Reviewer: ‘Il. 207-208: It would be helpful to show in a figure the salt bridges that are listed here.’

Response: As requested by the reviewer, we have revised Supplementary Fig. 8 to include the salt bridges listed.

Reviewer: ‘Il. 214-216: The paper that first described aerolysin rivet should be cited : 10.1038/sj.emboj.7600959 Iacovache et al. EMBO J 2006.’

Response: As requested by the reviewer, we have included the reference above in line 201, Ref. 44

Reviewer: ‘Il. 305-307. Reference 21 does not report that Etx Y84 and F105 are important for Etx function. The author should fix this and cite the correct reference.’

Response: We are very grateful for the reviewer to spot this mistake. We have replaced reference 21 with reference 47 in line 354 of the revised manuscript.

Reviewer: ‘Sup. Fig. 1. What is the band slightly below 75 kDa?’

Response: In the revised Supplementary Information file Sup. Fig. 1 now appears as Sup. Fig. 2. The faint band slightly below 75 kDa is likely to be the dimer of the faint band at 25 kDa, which likely corresponds to non-specific cleavage of recombinant wild type Etx protoxin by trypsin at a surface exposed arginine residue at position 227 based on sequence analysis using ExPASy-PeptideCutter. This by-product of trypsin activation can only be detected by immunoblotting using anti-His antibody and does not affect oligomer and pore formation.

Reviewer: ‘Sup. Fig. 2. The authors could box a few particles as they did in Fig. 2.’

Response: In the revised Supplementary Information file Sup. Fig. 2 now appears as Sup. Fig. 3. As requested by the reviewer, we boxed a few particles as we did in Fig. 2.

Reviewer: ‘Sup. Fig. 5 legend : should read « was used to collect» and not « was used top collect »

Response: In the revised Supplementary Information file Sup. Fig. 5 now appears as Sup. Fig. 6. As requested by the reviewer, we have revised the figure legend to remove the above mentioned mistake.

Reviewers' Comments:

Reviewer #1:

Remarks to the Author:

This manuscript was revised appropriately. This reviewer understands significance of the sample preparation and use of VPP on this study. In addition, additional experiments using H162A and D250A mutants, importance of interaction between H162 and D250 was revealed experimentally. Movie newly added also helps reader to understand the dynamic mechanism of EXT. This reviewer felt that this manuscript is worth for publication in Nat Commun. in terms of both scientific significance of clarifying dynamic mechanism of pore forming toxin and methodology of cryo-electron microscopy. The followings are several points to be revised before publication.

1. Authors emphasized the significance of sample preparation. However, the method used in this study seems to be same as that shown in Shimada et al., J. Biochem (2011). Please clearly describe what the difference is.
2. The author succeeded in improving resolution using VPP. Is this the first to improve resolution using VPP? If so, please clearly mention it. If there has been similar results, please refer it.
3. Line 229. Authors mentioned that the RBD in the oligomer overlaps well with that of monomer. Please show superposed structure as a figure. It should help readers to understand.
4. Line 217, 218.
ND1: D should be shown as delta in greek
OD2: D should be shown as delta in greek

Reviewer #2:

Remarks to the Author:

With their revised manuscript, the authors have satisfactorily addressed all my concerns. I therefore strongly recommend to accept their high quality manuscript, which will definitely be of interest to the broad readership of Nature Communications.

Reviewer #1

This manuscript was revised appropriately. This reviewer understands significance of the sample preparation and use of VPP on this study. In addition, additional experiments using H162A and D250A mutants, importance of interaction between H162 and D250 was revealed experimentally. Movie newly added also helps reader to understand the dynamic mechanism of EXT. This reviewer felt that this manuscript is worth for publication in Nat Commun. in terms of both scientific significance of clarifying dynamic mechanism of pore forming toxin and methodology of cryo-electron microscopy. The followings are several points to be revised before publication.

1. Authors emphasized the significance of sample preparation. However, the method used in this study seems to be same as that shown in Shimada et al., J. Biochem (2011). Please clearly describe what the difference is.
2. The author succeeded in improving resolution using VPP. Is this the first to improve resolution using VPP? If so, please clearly mention it. If there has been similar results, please refer it.
3. Line 229. Authors mentioned that the RBD in the oligomer overlaps well with that of monomer. Please show superposed structure as a figure. It should help readers to understand.
4. Line 217, 218.
ND1: D should be shown as delta in greek
OD2: D should be shown as delta in greek

Point-by-point response to Reviewer #1's comments

Reviewer: Authors emphasized the significance of sample preparation. However, the method used in this study seems to be same as that shown in Shimada et al., J. Biochem (2011). Please clearly describe what the difference is.

Response:

To assemble epsilon toxin pore complex, Shimada's study used MDCK cells, the most sensitive cell line known in 2011, while our study used Super Dome cells that form domes that are approximately five times the area of MDCK cells, thus exhibiting exceptional sensitivity to Etx, reported to have 3 to 78-fold increased sensitivity to Etx relative to MDCK cells. To solubilise the pore complex from cells, Shimada's study used solubilisation buffer containing 0.05% DDM, while our study used solubilisation buffer containing 2% DDM. The use of Super Dome cells, combined with the use of increased concentration of DDM maximised pore yield for subsequent purification, and thus were critical for structural determination of the pore complex.

To emphasize the significance of sample preparation, we have revised the Results section (lines 132-135) by replacing: 'We assembled Etx oligomers that were biologically active towards Super Dome cells, a clone of MDCK cells that exhibits exceptional sensitivity to Etx³⁹.' with:

'We assembled Etx oligomers that were biologically active towards Super Dome cells, a clone of MDCK cells that forms domes that are approximately five times the area of MDCK cells, thus exhibiting exceptional sensitivity to Etx³⁹.'

We have also revised the Methods section (lines 516-522) by replacing: ‘the pellet was resuspended in DPBS containing 137 mM NaCl and 2% (w/v) DDM and incubated overnight at 4 °C with occasional shaking.’ with:

‘Solubilisation of toxin complex from Super Dome cells was carried out by resuspending the pellet in DPBS (pH 7.0 – 7.2) containing 137 mM NaCl and 2% (w/v) n-dodecyl- β -D-maltoside (DDM) and incubation at 4 °C overnight, a modification of the method described by Shimada *et al.*³⁴ where the toxin complex from MDCK cells was solubilized by resuspending the pellet in 20 mM sodium phosphate buffer (pH 7.4) containing 0.05% (w/v) DDM and incubation at 25 °C for 1 h.’

In addition, we have revised the Methods section ‘Cell culture’ (lines 489-490) to include: ‘For dome formation, Super Dome cells were fed twice a day by a complete medium change.’

Finally, we have revised the Discussion section (lines 320-324) to include:

‘In this study, the use of Super Dome cells, reported to have 3 to 78-fold increased sensitivity to Etx relative to MDCK cells³⁹, combined with the use of increased concentration of DDM (2% (w/v)) relative to 0.05% (w/v) used in³⁴ maximized pore yield for subsequent purification, and thus were critical for structural determination of the Etx pore complex.’

Reviewer: The author succeeded in improving resolution using VPP. Is this the first to improve resolution using VPP? If so, please clearly mention it. If there has been similar results, please refer it.

Response: The VPP was developed to enable structural determination of small, <150 kDa complexes and has been shown in many studies to help in high-resolution structural determination of small complexes. Etx is not a particularly small complex, however the VPP also helped in our case. While our manuscript was under review another group published the 500 kDa structure of the human TFIIF core complex and also managed to improve the resolution from 4.3Å to 3.7Å with the use of the VPP. We have revised the Discussion section (lines 327-328) as follows:

‘Use of the VPP proved essential in our studies to achieve high resolution, owing to the challenging sample rather than its molecular mass (~225 kDa).’

We have also revised the Discussion section (lines 333-336) to include:

‘While this manuscript was under review, another cryo-EM study of a relatively large complex (~500 kDa) also benefited by the use of the VPP⁴⁷, suggesting that VPP use could be advantageous for certain specimens which are not size limited.’

Reviewer: Line 229. Authors mentioned that the RBD in the oligomer overlaps well with that of monomer. Please show superposed structure as a figure. It should help readers to understand.

Response: We have made a new supplementary figure (Supplementary Fig. 11) showing the superposition of the RBD in the water soluble monomer and the membrane-inserted form. Consequently, we have changed the numbering of subsequent Supplementary Figures.

Reviewer: Line 217, 218.

ND1: D should be shown as delta in greek

OD2: D should be shown as delta in greek

Response: We have corrected this (lines 224-225) and we thank the reviewer.